# The Bi-(AID-1-T) G-Quadruplex Has a Janus Effect on Primary and Recurrent Gliomas: Anti-Proliferation and Pro-Migration

**DOI:** 10.3390/ph17010074

**Published:** 2024-01-07

**Authors:** Svetlana Pavlova, Lika Fab, Ekaterina Savchenko, Anastasia Ryabova, Marina Ryzhova, Alexander Revishchin, Igor Pronin, Dmitry Usachev, Alexey Kopylov, Galina Pavlova

**Affiliations:** 1Institute of Higher Nervous Activity and Neurophysiology, Russian Academy of Sciences, 117485 Moscow, Russia; 2Federal State Autonomous Institution N. N. Burdenko National Medical Research Center of Neurosurgery of the Ministry of Health of the Russian Federation, 125047 Moscow, Russia; 3Prokhorov General Physics Institute of the Russian Academy of Sciences, 119991 Moscow, Russia; 4Belozersky Research Institute of Physical Chemical Biology, Lomonosov Moscow State University, 119991 Moscow, Russia; kopylov.alex@gmail.com; 5Department of Medical Genetics, Sechenov First Moscow State Medical University, 119991 Moscow, Russia

**Keywords:** glioma, migration, G-quadruplexes, anti-proliferative activity, treatment, resistance

## Abstract

High-grade gliomas are considered an incurable disease. Despite all the various therapy options available, patient survival remains low, and the tumor usually returns. Tumor resistance to conventional therapy and stimulation of the migratory activity of surviving cells are the main factors that lead to recurrent tumors. When developing new treatment approaches, the effect is most often evaluated on standard and phenotypically depleted cancer cell lines. Moreover, there is much focus on the anti-proliferative effect of such therapies without considering the possible stimulation of migratory activity. In this paper, we studied how glioma cell migration changes after exposure to bi-(AID-1-T), an anti-proliferative aptamer. We investigated the effect of this aptamer on eight human glioma cell cultures (Grades III and IV) that were derived from patients’ tumor tissue; the difference between primary and recurrent tumors was taken into account. Despite its strong anti-proliferative activity, bi-(AID-1-T) was shown to induce migration of recurrent tumor cells. This result shows the importance of studying the effect of therapeutic molecules on the invasive properties of glioma tumor cells in order to reduce the likelihood of inducing tumor recurrence.

## 1. Introduction

Therapeutic strategies for aggressive cancers like glioma continue to evolve rapidly. Gliomas, classified as primary malignant brain tumors originating from neuroectodermal tissue, are notably different from other cancer types, possessing specific characteristics. For instance, while gliomas do not metastasize, they can invade surrounding tissue. This means that glioma cells can migrate within the brain, leading to the development of recurrent tumors at distant sites from the primary tumor.

There are many features by which gliomas can be classified, but some of the main ones are considered to be classified by grade of malignancy (Grade I–IV) and by the presence of a mutation in the IDH1 gene (IDH1− and IDH1+), where IDH1− is also called “wild type” and IDH1+ is called “mutant type”. Glioblastoma (GBM) is the most aggressive of the glial tumors. It is classified as a Grade IV brain tumor under the 2021 WHO Classification of Tumors of the Central Nervous System. GBM patients have the lowest survival rate, and their predicted life expectancy is usually less than two years [1,2].

Temozolomide (TMZ) is currently a leading drug in glioma treatment, but its results are inconsistent. While it decreases cell proliferation in some glioma cells, it paradoxically increases it in others, simultaneously enhancing tumor cell migration. Both outcomes are unfavorable [3]. Radiotherapy, too, has mixed results, often sparing proliferative and migratory cells that subsequently evade the treatment zone [4]. Attempts to combine tumor surgery, radiation therapy, and chemotherapy have only managed to slow tumor growth temporarily, often resulting in drug resistance and therapy complications [5,6]. Thus, there is a disappointing pattern indicating that known therapeutic approaches, which can slow down tumor growth, do induce recurrent tumors via the migration of tumor cells. Meanwhile, tumor growth retardation also turns out to be short-lived; under any influence, the growth is eventually restored.

The need to search for new approaches to the treatment of gliomas becomes obvious.

Aptamers seem attractive for use as therapeutic agents because they meet many criteria. Aptamers are a product of chemical synthesis hence their purity can be controlled. Similar to antibodies, aptamers bind to their targets with high affinity. However, they are smaller in size compared to antibodies, thus allowing improved transport to the target [7]. The small size of aptamers enables them to cross the blood–brain barrier, which is essential for glioma therapy [8].

G-quadruplexes (GQs) are special aptamers and lately, they have been widely considered antitumor molecules. These guanine-rich GQ oligonucleotides are characterized by increased thermodynamic stability and reduced immunogenicity—properties that promote more precise interaction with the target protein [9,10].

Bi-modular GQ, bi-(AID-1-T), has been shown to inhibit glioma cell proliferation [11]. It was synthesized by covalent conjugation of two AID-1-T molecules via three thymidine links, which was shown not to affect its GQ structure [12]. Moreover, bi-(AID-1-T) has no effect on normal cells (human fibroblasts), thereby making it more favorable as a proliferation-reducing factor for glioma treatment compared to other GQ oligonucleotides such as AID-1-T, HD1, bi-HD1, and AS1411 [11].

This work is aimed at further studying the anti-proliferative effect of bi-(AID-1-T) on cell cultures. We focused our attention on primary glioma cultures obtained from surgical specimens of patients. All the investigated cultures were divided into two groups based on the type of initial tumor—primary gliomas and recurrent gliomas. In addition, we evaluated the effect of bi-(AID-1-T) on the migratory activity of these cell cultures.

## 2. Results

Eight human glioma cell cultures obtained from surgical specimens of patients were used in this work: four primary glioma cultures and four secondary glioma cultures (recurrences). A more detailed description is presented in Table 1.

Tumors developing from a recurrent primary tumor are known to be highly resistant to further therapy. For this reason, we studied the effects of aptamers on these two groups of cell cultures separately.

Unlike aptamers whose target molecules are known, Kopylov A. et al. introduced a new term called “crypto-aptamer”, describing an aptamer whose protein target is currently unknown [11,13]. It is assumed that when the target protein is identified, it becomes an aptamer.

In the first stage of the research, the effects of the G-quadruplex AID-1-T (15 bp) and its dimer bi-(AID-1-T) (33 bp) on human glioma cell cultures were evaluated. One culture (Sus\fP2) from the primary tumor group and two cultures (G-01 and G-23) from the recurrent group were selected for this purpose.

To evaluate the proliferative activity of the cell cultures, an MTS assay was performed 72 h after the addition of the aptamers at different concentrations.

Results differed for the selected cell cultures and crypto-aptamers. The Sus\fP2 cell culture was most affected, the G-23 cell culture was moderately affected, while in the G-01 cell culture, AID-1-T application had almost no anti-proliferative effect (Figure 1A). Treatment with bi-(AID-1-T) dramatically reduced cell proliferation in Sus\fP2 and G-23 cell cultures (Figure 1B); however, no significant results were obtained for G-01. These experiments demonstrate that the anti-proliferative effect was optimal when the aptamer was added at a 10 μM concentration. This value was found to be appropriate for further studies.

In addition, cell viability was assessed by flow cytofluorometry. Sus\fP2 culture was selected for this purpose, which showed the most pronounced effect.

The treated culture had a significant increase in the proportion of apoptotic cells compared to the culture not exposed to GQs. But, the most pronounced increase in cell count in early and late apoptosis was observed after the addition of bi-(AID-1-T) (Figure 1C,D).

The expression levels of genes associated with stemness and malignancy after exposure to the crypto-aptamers were assessed by immunocytochemistry (ICC). It is important to note that G-01 can be considered the least sensitive cell culture, while Sus\fP2 is the most sensitive to the crypto-aptamers. Seventy-two hours after AID-1-T treatment, L1CAM and p53 levels decreased, and Nestin levels increased in Sus\fP2. Nestin and Sox2 were also upregulated in G-01, while Casp3 expression was downregulated in this cell culture (Figure 1E). After the addition of bi-(AID-1-T), Nestin expression level in the Sus\fP2 culture increased, whereas decreased p53 expression level and increased Sox2 and CD44 expression levels were observed in the G-01 culture (Figure 1F). This likely indicates the presence of different stem cell populations.

Based on these findings, we selected bi-(AID-1-T) as the most suitable G-quadruplex for further studies.

Before expanding the cell culture panel to evaluate the effect of GQ on cell cultures, we investigated the rate of penetration of the bi-(AID-1-T) oligonucleotide into G-01 glioblastoma cells, as well as its localization 1.5, 3, 24, and 72 h after treatment. An additional fragment at the 3′-end of the bi-(AID-1-T) DNA crypto-aptamer, complementary to the FAM-labeled DNA, was used to introduce the fluorescence label.

Bi-(AID-1-T)-FAM penetrated the cells within 1.5 h already and accumulated up to 24 h. By 72 h, there was a gradual decrease in its intracellular concentration or disintegration. The aptamer was evenly distributed in the cell cytoplasm and accumulated in the perinuclear space (Figure 2).

Using an expanded cell culture panel, which included four cell cultures derived from primary gliomas and four cell cultures derived from recurrent gliomas, we evaluated the anti-proliferative effect of bi-(AID-1-T) at a 10 μM concentration.

It predominantly affected the cell cultures derived from primary tumors (Figure 3A). At the same time, cell viability test results were more significant for cell cultures derived from recurrent tumors (Figure 3B,C).

Immunocytochemical (ICC) analysis showed an upregulation of CD133, L1CAM, Casp3, Sox2, and CD44 in cell cultures derived from primary tumors after bi-(AID-1-T) had been added to the unexposed controls (Figure 3D). In addition to increased expression levels of L1CAM and CD44, cell cultures from recurrent gliomas also had increased expression levels of EGFR and Nestin, while p53 was downregulated (Figure 3E).

When we evaluated the migratory activity after treatment with bi-(AID-1-T), we observed a general trend toward a decrease in cell cultures derived from primary gliomas, except for the Sus\fP2 cell culture. The migration rate increased in this cell culture. At the same time, in most cell cultures derived from recurrent tumors, the migratory activity was stimulated by bi-(AID-1-T) (Figure 3F).

## 3. Discussion

Gliomas, and particularly the highly malignant ones, represent a huge challenge for mankind. Despite the existence of various approaches to its treatment (surgical removal of the tumor, chemotherapy, and radiotherapy), the patient survival rate remains very low. It has been found that an important feature of gliomas is that some tumor cells can resist radiotherapy and chemotherapy [14,15]. In response to therapeutic intervention, these resistant cells begin to migrate out of the tumor site, leading to a recurrent tumor in another brain region. That is, one of the main factors of recurrent glioma is the invasive and migratory abilities of tumor cells.

Surprisingly, glioma cells surviving after therapy escape from the tumor to the parenchyma of a healthy human brain like living beings. It seems that the more intense and effective the therapy is, the higher the ability of surviving cells to migrate [16,17,18]. For example, Shimizu et al. showed that bevacizumab therapy kills many tumor cells, but surviving cells have upregulation of δ-catenin expression, and it is associated with increased invasion of glioma cells [15]. Another study by Kochanowski et al. showed that TMZ possibly enhances the invasiveness of surviving tumor cells and that this invasiveness correlates with upregulation of ectopic expression of Snail1 [17].

There have been similar reports on the migratory or invasive properties of tumor cells after radiation therapy. Researchers have reported observing increased migratory/invasive activity of surviving cells after radiation exposure [16,19,20]. Radiotherapy has been shown to be associated with upregulation of genes involved in proliferation and migration of tumor cells (CXCL12, VEGF-A, TGF-β1, and TNFα). Cells that have similar properties and tend to escape the zone of death are more likely to survive [4].

A combination of different therapy options (chemotherapy and radiation therapy) was found to slow down tumor growth temporarily; however, surviving tumor cells exhibited enhanced growth and became resistant to further therapy [5,6].

It has been suggested that therapy narrows the clonal diversity of tumors due to the survival of the most resistant clones; this makes recurrent tumors even more resistant to subsequent therapy [5,21]. It is important to note that several studies have reported that recurrent tumors are typically more aggressive [22] and have increased migratory ability due to mutations induced by chemotherapy and radiotherapy [4,6,23] or due to the survival of tumor stem cells with enhanced migratory properties [24].

Apparently, when developing new therapeutic modalities, it is necessary to evaluate the migration of tumor cells after effectively inhibiting their proliferation. The point is that upon therapeutic downregulation of cancer cell proliferation, in particular, of cancer stem cells, a “protective” mechanism to “rescue” the aggressive tumor cells is activated, and they escape from the zone of death. So, this work was aimed at evaluating the migration properties of human glioma tumor cells after exposure to aptamers that have anti-proliferative properties [11].

AID-1-T and its more stable dimer bi-(AID-1-T) [24], which showed anti-proliferative effect against tumor cell lines in the work by Legatova et al. [11], were used as the G-quadruplexes under study.

In a comparative evaluation of the anti-proliferative properties of AID-1-T and bi-(AID-1-T) on three glioma cell cultures, bi-(AID-1-T) was shown to more effectively reduce the ability of glioma tumor cells to divide. It should be pointed out that bi-(AID-1-T) had a dose-dependent efficacy. On the most sensitive glioma cell culture Sus\fP2, there was increased apoptosis, with virtually no cell necrosis, after exposure to the aptamer.

Bi-(AID-1-T) has been shown to downregulate Casp3 and p53 expressions significantly. High Casp3 expression levels are usually associated with higher tumor resistance and poorer patient survival. Casp3 levels were found to correlate with tumor angiogenesis and proliferation markers [25]. p53 is an important regulator of apoptosis and cell proliferation processes. Low p53 expression levels are considered a favorable factor in predicting survival [26,27]. That is, after exposure to bi-(AID-1-T), the cells that are upregulating Casp3 and p53 expressions die, and this may have a positive effect on tumor prognosis. However, we found that Sox2 representation is increased in aptamer-resistant tumor cells, indicating the survival of a certain group of tumor stem cells.

Using fluorescently labeled bi-(AID-1-T)-FAM, it was shown that the aptamer actively penetrates tumor cells as early as one hour after it has been added to the medium, followed by gradual accumulation in the cytoplasm. Optimal accumulation of the aptamer occurs after 24 h of incubation. Moreover, the GQ is intensively presented in cells 72 h after addition (Figure 2). Thus, a time interval of 72 h between GQ exposure and tumor cell analysis was chosen for further studies.

Eight primary cultures of high-grade gliomas (Garde III, IV) obtained from surgical specimens of patients were analyzed. The cultures were grouped based on the type of the original tumor: group 1 (primary tumors) and group 2 (recurrent tumor). All cell cultures were IDH1-.

Bi-(AID-1-T) had its greatest anti-proliferative effect on primary glioma tumor cultures, while the cultures obtained from recurrent tumors remained fairly resistant to treatment. L1CAM- and CD44-positive cells were the surviving ones. In addition, Sox2 expression was observed in primary tumor cells, while Nestin was expressed in recurrent tumors. This indicates the survival of predominantly tumor stem cells. However, it can be suggested that there may be types or subpopulations of stem cells with different Sox2 and Nestin levels.

Several studies have suggested that a high percentage of Nestin-positive cells correlates with aggressiveness and poor prognosis [28]; however, such observations are found in homogenous glioma cell lines. Using glioma cell cultures, we noted increased Nestin expression only in the surviving tumor cells of recurrent tumors, whereas primary tumors had no such change after exposure to the aptamer. This result requires further investigation. However, it is clear from the results we obtained that tumor cell proliferation significantly decreases after exposure to bi-(AID-1-T) in primary cell cultures, with recurrent tumors being more resistant to such effects. By reducing their proliferative activity, the surviving tumor cells show greater migratory ability. Moreover, upregulation of L1CAM and CD44 expression in surviving glioma tumor cells that are detected after aptamer exposure is often associated with increased cell migration ability. These characteristics are usually attributed to tumor stem cells [29,30,31,32].

The use of aptamers and G-quadruplexes in general for anti-cancer therapy is becoming very popular [33,34]. The use of aptamers also seems to be the future. However, one should note that the effect on tumor cells may induce their migratory activity and, consequently, a recurrent tumor. Therefore, apart from its anti-proliferative features, the migration-inhibiting properties of an aptamer for anti-cancer therapy should be analyzed.

We did not find any studies that show that while GQs have an effective anti-proliferative effect on tumor cells, they nevertheless stimulate the migration of surviving cells. On the contrary, it has been argued that GQs reduce cell migratory activity [33,35,36]. Such results may be down to the fact that the effect of GQs on tumor cell migratory activity is mostly considered only on tumor cell lines and not on heterogeneous cell cultures derived from patient tumor tissue. Only one study has demonstrated that in the A549 line, G-quadruplex TMPyP4 was shown to promote tumor cell migration [37].

We recognize that given the high variability in glioma types and grades, we used a rather small sample. Also, we did not consider in our studies the status of p53 mutations, which may also have an effect on culture sensitivity. However, despite this, we report the disturbing result that bi-(AID-1-T) “without mediators” and “companions”, although confirmed to have anti-proliferative and cytostatic effects, is able to induce an enhanced migration activity of human glioma cells. This indicates its dual nature or the so-called Janus effect, which has been previously mentioned in articles [38] and which cannot be ignored in further research.

## 4. Materials and Methods

### 4.1. Cell Cultures

Primary cultures were obtained from human postoperative glioma, complying with all the formal requirements of the Russian Federation. This study was approved by the Ethics Committee of Burdenko Neurosurgical Institute, Russian Academy of Medical Sciences (No_12/2020). All subjects gave written informed consent in accordance with the Declaration of Helsinki. All cell cultures were adherent cell cultures. The cells were cultured in DMEM/F12 medium (Gibco, Thermo Fisher Scientific, Carlsbad, CA, USA) supplemented with 10% FBS (Gibco, USA), 2 mM L-glutamic acid (Paneco, Moscow, Russia), and 1% antibiotic/antimycotic solution (penicillin/streptomycin) (Corning, Corning, NY, USA) at 37 °С and 5% СО_2_. Cells were removed from culture vessels using Versene solution (Paneco, Russia) and 0.05% Trypsin solution (Gibco, USA).

### 4.2. Aptamer Treatment

The aptamers were treated before use: 100 µM aptamer solution was incubated at 95 °C for 5 min in a buffer (10 mM KCl, 140 mM NaCl, 20 mM Tris-HCl). They were then cooled at room temperature. After that, solutions of required concentrations (10, 20, and 30 µM) were prepared and added to cell cultures for a 72 h incubation. Controls were supplemented with equivalent amounts of buffer instead of the aptamer.

### 4.3. MTS Assay

Ninety-six-well TC-treated microplates (Corning, USA) were planted with 4000 cells/well. Cells were counted by TC20 Automated Cell Counter (Bio-Rad, Hercules, CA, USA). Cells were incubated for 72 h, after which 1/10 of the medium volume of MTS reagent (Promega, Madison, WI, USA) was added to each well. After 90 min of incubation at 37 °С and 5% СО_2_, the absorbance was measured by SPECTROstar Nano (BMG LABTECH, Ortenberg, Germany) at 495 nm.

### 4.4. Flow Cytometry

To identify apoptotic and necrotic cells, we used the FITC Annexin V Apoptosis Detection Kit with PI (BioLegend, London, UK). The cells were treated according to the manufacturer’s guidelines. Upon the addition of 200 µL of the buffer for Annexin V binding, each sample was analyzed on BD FACSAria^TM^III (BD Bioscience, Franklin Lakes, NJ, USA) using FITC-A and PE-A detection. A total of 30,000 events were analyzed with the supplied software BD FACSDiva v8.0.2 (BD Bioscience, USA). Flow cytometry data were analyzed using FlowJo v10.6.2 (FlowJo LCC, Franklin Lakes, NJ, USA).

### 4.5. Immunocytochemistry (ICC)

Cells were planted into culture plates with predisposed Coverslips 20009 (SPL Life Sciences, Gyeonggi-do, Korea) (2 × 10^4^ cells per coverslip) and incubated for 24 h at 37 °С and 5% СО_2_. After the aptamer had been added, the cells were incubated for another 72 h. After that, the cells were fixed with 4% paraformaldehyde and stained with primary antibodies for L1СAM (MA-14-140, Thermo Fisher Scientific, Carlsbad, CA, USA), Sox2 (sc-17320, Santa Cruz Biotechnology, Dallas, TX, USA), p53 (Ab1101, Abcam, Cambrige, UK), CD44 (14-044-082, Invitrogen, Thermo Fisher Scientific, Carlsbad, CA, USA), Casp3 (Ab2303, Abcam, Cambrige, UK), and secondary antibodies labeled with Cy2 (715-225-151, Jackson ImmunoResearch, UK) or Alexa Fluor^®^ 594 (711-585-152, Jackson ImmunoResearch, West Grove, PA, USA). Cell nuclei were stained with Bisbenzimide Hoechst 33342 (Sigma-Aldrich, Burlington, MA, USA) dissolved in PBS. Cells were fixed in Mowiol 4–88 (9002-89-5, Sigma-Aldrich, Burlington, MA, USA) with 1% DABCO for 8 h at 4 °С and then for 24 h at room temperature. The samples were analyzed using a Carl Zeiss confocal microscope of LSM-710 series. Cell cultures stained with secondary antibodies without primary antibodies were used as antibody controls. The average fluorescence intensity of cells was determined using the ImageJ v1.53n software.

### 4.6. Assessment of Cell Penetration and Distribution

Aptamer distribution in cells was assessed using apto-cytochemistry. Coverslips (SPL Life Sciences, Korea) with cells were prepared as described above (Section 4.5), and then 1 µM of FAM-bi-(AID-1-T) was added. After 1.5, 3, 24, and 72 h of incubation, the cells were fixed with 4% paraformaldehyde, stained with Bisbenzimide Hoechst 33342 (Sigma, USA) dissolved in PBS, and analyzed using LSM-710 Carl Zeiss confocal microscope.

### 4.7. Assessment of Cell Migratory Activity

Migration activity was assessed using Transwell inserts in 24-well plates with 8 µm pores (353093, Corning, USA). Cells were incubated in a serum-free medium for 1 h and then planted in 300 µL of FBS-free medium (10^4^ cells per well) and supplemented with 700 µL of full medium (Section 4.1). After 20 h, the inserts and the non-migrated cells were removed. The migrated cells were fixed with 4% paraformaldehyde and stained with Bisbenzimide Hoechst 33342 (Sigma, USA). Migratory activity was defined by cell count in 20 non-crossing visual fields.

### 4.8. Statistics

Statistical analysis (ANOVA test) was reformed with GraphPad Prism 9 software. The data are presented as means ± SD from a minimum of 3 repeats. Statistical significance is presented as * *p* < 0.05, ** *p* < 0.01, *** *p* < 0.001 and **** *p* < 0.0001.

## Figures and Tables

**Figure 1 pharmaceuticals-17-00074-f001:**
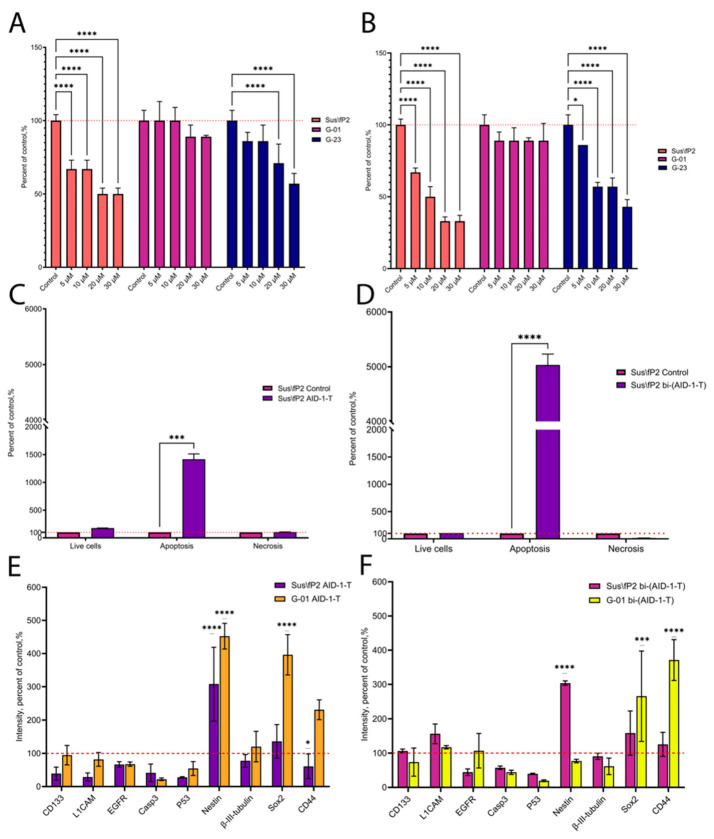
Effect of G-quadruplexes (GQs) AID-1-T and its dimer bi-(AID-1-T) on Sus\fP2 (primary glioma), G-01, and G-23 (recurrent glioma) cell cultures. (**A**) Proliferative activity estimation (MTS assay) in Sus\fP2, G-01, and G-23 cultures when treated with AID-1-T at 5 μM, 10 μM, 20 μM, and 30 μM concentrations; data are presented as a percentage of the control, which was the corresponding non-treated cell culture. (**B**) Proliferative activity estimation (MTS assay) in Sus\fP2, G-01, and G-23 when treated with bi-(AID-1-T) at 5 μM, 10 μM, 20 μM, and 30 μM concentrations; data are presented as a percentage of the control, which was the corresponding non-treated cell culture. (**C**) Cell viability assessment (flow cytometry) in Sus\fP2 after treatment with 10 μM AID-1-T; data are presented as a percentage of the control, which was the corresponding non-treated cell culture. (**D**) Cell viability assessment (flow cytometry) in Sus\fP2 after treatment with 10 μM bi-(AID-1-T); data are presented as a percentage of the control, which was the corresponding non-treated cell culture. (**E**) immunocytochemical staining (ICC) of Sus\fP2 and G-01 cell cultures after treatment with 10 μM AID-1-T. (**F**) ICC staining of Sus\fP2 and G-01 cell cultures after treatment with 10 μM bi-(AID-1-T). CD133 (prominin-1)—stem cells marker; L1CAM—L1 cell adhesion molecule, stem cells, and adhesion marker; EGFR—epidermal growth factor receptor, malignancy marker; Casp3—caspase-3, apoptosis marker; p53—malignancy marker; Nestin-stem cells marker, β-III-tubulin-neuroblast marker, Sox2-stem cells marker, CD44-stem cells and migration marker. * *p* < 0.05, *** *p* < 0.001 and **** *p* < 0.0001. The red dashed line indicates the level of controls taken as 100%.

**Figure 2 pharmaceuticals-17-00074-f002:**
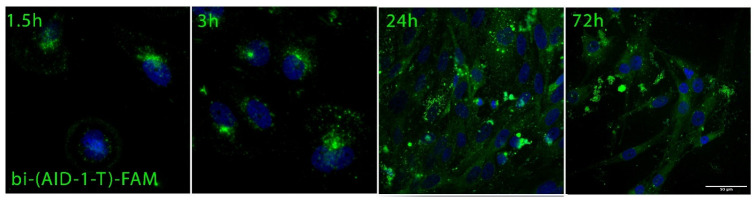
Estimation of FAM-labeled bi-(AID-1-T) localization in G-01 cell culture 1.5, 3, 24, and 72 h after treatment with the cryo-aptamer at a 1 μM concentration. Scale bar 50 µm.

**Figure 3 pharmaceuticals-17-00074-f003:**
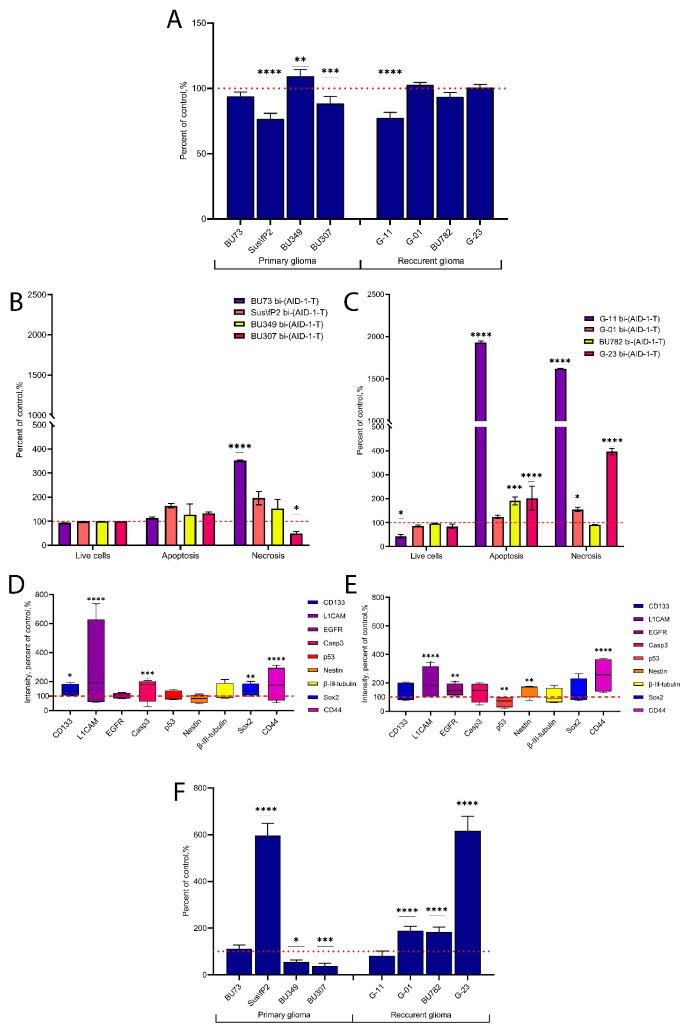
Effect of G-quadruplex-based dimeric aptamer bi-(AID-1-T) (10 μM) on cell cultures derived from primary gliomas and cell cultures from recurrent gliomas. (**A**) Proliferative activity estimation (MTS assay) in cell cultures treated with bi-(AID-1-T) at a 10 μM concentration; data are presented as a percentage of the control, which was the corresponding non-treated cell culture. (**B**) Cell viability assessment (flow cytometry) in cell cultures derived from primary gliomas after treatment with 10 μM bi-(AID-1-T); data are presented as a percentage of the control, which was the corresponding non-treated cell culture. (**C**) Cell viability assessment (flow cytometry) in cell cultures derived from recurrent gliomas after treatment with 10 μM bi-(AID-1-T); data are presented as a percentage of the control, which was the corresponding non-treated cell culture. (**D**) ICC staining of cell cultures derived from primary gliomas after treatment with 10 μM bi-(AID-1-T); data are presented as a percentage of the control, which was the corresponding non-treated cell culture. (**E**) ICC staining of cell cultures derived from recurrent gliomas after treatment with 10 μM bi-(AID-1-T); data are presented as a percentage of the control, which was the corresponding non-treated cell culture. CD133 (prominin-1)—stem cells marker; L1CAM—L1 cell adhesion molecule, stem cells, and adhesion marker; EGFR—epidermal growth factor receptor, malignancy marker; Casp3-caspase-3, apoptosis marker; p53-malignancy marker; Nestin-stem cells marker, β-III-tubulin-neuroblast marker, Sox2-stem cells marker, CD44-stem cells and migration marker. (**F**) Migration estimation in cell cultures after treatment with 10 μM bi-(AID-1-T). * *p* < 0.05, ** *p* < 0.01, *** *p* < 0.001 and **** *p* < 0.0001. The red dashed line indicates the level of controls taken as 100%.

**Table 1 pharmaceuticals-17-00074-t001:** List of primary human glioma cell cultures obtained from surgical specimens of patients with a brief description of tumors.

	Name	Grade	IDH1 Status	Tumor Location
Primary glioma	BU73	III	wild type	Right temporal lobe
	Sus\fP2	IV	wild type	Left temporal lobe
	BU349	IV	wild type	Left frontal lobe
	BU307	IV	wild type	Left temporal lobe
Recurrent glioma	G-11	III-IV	wild type	Right frontal–temporal–insular region
	G-01	IV	wild type	Left frontal lobe
	BU782	IV	wild type	Parieto-occipital region
	G-23	IV	wild type	Left frontal parasagittal region

## Data Availability

The data presented in this study are available upon request from the corresponding author. The data are not publicly available due to the reason that some of the generated data are not published.

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
