# Peer review of "The Bi-(AID-1-T) G-Quadruplex Has a Janus Effect on Primary and Recurrent Gliomas: Anti-Proliferation and Pro-Migration"

_pharmaceuticals, 2024, doi:10.3390/ph17010074_

Round 1

Reviewer 1 Report

Comments and Suggestions for Authors

While this is an interesting and important topic, but this manuscript is fairly difficult to rad due to multiple grammar and syntax problems.  For example, line 43: replace ‘By now’ with As of now; see also lines 55, 65, 94, 117  (replace ‘past’ with ‘after’; note that is same comment applies to many places in the manuscript), 114, 166, 204, 207, 217, 229,247, 261 264, 283 for various word choice issues.  In line 25, what does ‘it’ refer to?  Also, avoid starting some sentences with a numeral; rather write the word (for example see line 249  and replace 8 with ‘eight’).

In addition, there are too many undefined abbreviations used through the manuscript.  For example, in Table 1, what does IDH1 status refer to?  The authors give the reader no context for this. 

For lines 155-157:  1 sentence is not really equal to 1 paragraph

For line 90,  the authors should add the word ‘for’  to read ‘….for which target molecules….’

The authors also made a number of statements that are not referenced:  for example in the paragraph (starting with line 58) in line 62 they indicate that the oligonucleotides are much smaller but do not give a range for what it meant by ‘smaller’.  What is the molecular weight of the bi(AID-1-T)?   Line 146 should have a reference for the statement ‘…since it was shown that…’

For Figure 1, line 137: the authors should define ‘ICC’ staining

For Figure 2, the authors give no rational for why 1 uM of FAM-labeled bi-AID-1-T) was used here.

Line 166: the start of this sentence is confusing; line 169; the phrase ‘..as it was demonstrated before..” should be referenced or show data.

For Figure 3, the authors should, in the text about this figure indicate what each abbreviation means in terms of protein name and biological activity.

 Line 178:  does this imply that the cells are not adherent cells in culture…the authors should indicate if the cells cultures originated from gliomas are adherent or not; if adherent, how were they released from the culture plates.

Line 246:  how do the authors define “intensely”?  Also, see Figure 2 where authors have indicated a decline. 

Line 251:  what is it important that all cell cultures were IDHI-?

In may places in this manuscript, the experimental design, controls and results are not clearly presented.  For example, how is ‘ICC’ staining done and what controls were used.  There were no references used in the Materials and Methods section and insufficient information presented if one would like to reproduce the data;  How long after obtaining the primary cultures were they used; were the primary cultures stored (frozen?), or used ‘fresh’?  How were the primary cultures assessed for cell population density prior to experiments with test compounds?  What is the percent of stem cells and how did this percent change?  Are these adherent cells?  In section 4.2  how was the ‘success’ of the aptamer preformation evaluated?  In section 4.3   what 96-well plate type was used and from what company were they obtained?  How were cells counted (since authors report they seeded 4000/well).

Line 315 :  data not shown? 

Line 317:  what does ‘predisposed coverslips’ mean?

Line 329: give reference for apto-cytochemistry method or more detail.

Line 333:  what were the authors looking for when using the confocal microscopy?  They do not report how many cells were counted by microscopy.

Line 340: how many cells per cell (% of number plates) were counted to give a ‘representative’ value?

Lines 272-275: should this be in the introduction rather than here.

Line 281:  what is TMPyP4?

Comments on the Quality of English Language

As indicated, there are many issues with the quality of the writing and use of undefined abbreviations 

Author Response

Dear reviewer, 
Thank you very much for taking the time to review this manuscript so carefully. I have carefully read all your comments and suggestions and I am providing a detailed response to them in the attachment.

Reviewer 2 Report

Comments and Suggestions for Authors

The authors present Bi-(AID-1-T) G-quadruplex has Janus effect on primary and  recurrent gliomas: anti-proliferation and pro-migration

The paper looks interesting, however I wanted to follow up on what a Janus effect is?

The majority of the paper is spent describing the phenotype of these cell lines. 

Important points. Do the authors have any IDH + cell lines?

Also please define how cells are undergoing apoptosis. Have the authors looked at Caspase-3 activation?

Is there a dose response of your G-quadruplex treatment with respect to anti-proliferation or pro-apoptosis?

Have the authors looked at migration with these dose cures as well?

Comments on the Quality of English Language

n/a

Author Response

Dear reviewer,

I thank you for reading my article so thoroughly. I am sure that by correcting the aspects you have pointed out, I will make our article better. I have attentively read all your comments and suggestions and a detailed response to them is given below.

  • The paper looks interesting, however I wanted to follow up on what a Janus effect is?

We decided to make a reference to the ancient Roman two-faced god Janus in the title of the article. In this case, by the Janus effect we mean the dual nature of aptamer's effects. On the one hand it helps in the treatment of gliomas by reducing the proliferation of tumor cells, but on the other hand it promotes the development of recurrences by stimulating the migration activity of tumor cells. The Janus effect (or a “Janus face”) is sometimes mentioned in articles, e.g. (Esposito, Veronica, Annapina Russo, Teresa Amato, Valentina Vellecco, Mariarosaria Bucci, Luciano Mayol, Giulia Russo, Antonella Virgilio, and Aldo Galeone. 2018. “The ‘Janus Face’ of the Thrombin Binding Aptamer: Investigating the Anticoagulant and Antiproliferative Properties through Straightforward Chemical Modifications.” Bioorganic Chemistry 76 (February): 202–9. https://doi.org/10.1016/j.bioorg.2017.11.005.)

  • The majority of the paper is spent describing the phenotype of these cell lines.

Since we are working with primary cell cultures derived from patients rather than cell lines, we cannot select a completely homogeneous collection of cell cultures. This diversity may affect the results of the studies. Therefore, we consider it is important to describe the cell cultures in such detail.

  • Do the authors have any IDH + cell lines?

Yes, we have IDH1+ cultures in our cell culture collection. But according to the literature IDH1- cultures are considered to be more malignant, they are less curable and have a worse prognosis for patients. For this reason, at the initial stage of the study we focused our attention on IDH1- glioma cell cultures. But in the future we plan to expand our study and repeat the experiments on IDH1+ cultures.

  • Have the authors looked at Caspase-3 activation?

We are very grateful to you for bringing this to our attention. We have an error in the captions of the immunocytochemical staining plots in Figures 1 and 2. It should indicate Casp3 instead of Cas3. We will correct this error for sure.

In the case of primary gliomas, we do see a statistically significant increase in caspase-3 expression levels. In the case of recurrent gliomas, we also observe an increase in the level of caspase-3 expression, but it is not statistically significant.

  • Is there a dose response of your G-quadruplex treatment with respect to anti-proliferation or pro-apoptosis?

We tested the effect of the aptamer on proliferative activity at different concentrations both in this work (Figure 1A) and in previous studies (Legatova, V. ; Samoylenkova, N.; Arutyunyan, A.; Tashlitsky, V.; Zavyalova, E.; Usachev, D.; Pavlova, G.; Kopylov, A. Covalent Bi-Modular Parallel and Antiparallel G-Quadruplex DNA Nanocostructs Reduce Viability of Patient Glioma Primary Cell Cultures. Int. J. Mol. Sci. 2021, 22, 3372. https://doi.org/10.3390/ijms22073372)

In the course of these studies, we chose the concentration of 10 μM as the most optimal and further experiments were carried out on it.

  • Have the authors looked at migration with these dose cures as well?

We are currently in the process of carrying out experiments to extend the range of concentrations to evaluate their effect on the migration activity of the cultures, which will probably be described in the next article

Reviewer 3 Report

Comments and Suggestions for Authors

The authors investigate the Bi-(AID-1-T) G-quadruplex that exhibits a Janus effect on primary and recurrent gliomas, displaying both anti-proliferative and pro-migratory properties. In the context of primary gliomas, the Bi-(AID-1-T) G-quadruplex was found to demonstrate an anti-proliferative effect, i.e., it could inhibit or slow down the growth and division of tumor cells, potentially impeding the progression of the primary tumor. However, in the case of recurrent gliomas, the Bi-(AID-1-T) G-quadruplex exhibited a pro-migratory effect. This implies that it may promote the migration of glioma cells, allowing them to spread to other areas or develop secondary tumors. The presented research is needed  in order to further understand the underlying mechanisms and potential therapeutic implications of these contrasting effects.

I have suggestion for improvement of the body text in order to make it more readable, especially in the introductory part:

1.       The first paragraph could be rewritten, for example in the following way: “Therapeutic strategies for aggressive cancers like glioma continue to evolve rapidly. Gliomas, classified as primary malignant brain tumors originating from neuroectodermal tissue, are notably different from other cancer types, possessing specific characteristics. For instance, while gliomas do not metastasize, they can invade surrounding tissue. This means that glioma cells can migrate within the brain, leading to the development of recurrent tumors at distant sites from the primary tumor.”

2.       The third paragraph at row 43 may sound better in the following form: “Temozolomide (TMZ) is currently a leading drug in glioma treatment, but its results are inconsistent. While it decreases cell proliferation in some glioma cells, it paradoxically increases it in others, simultaneously enhancing tumor cell migration. Both outcomes are unfavorable. Radiotherapy, too, has mixed results, often sparing proliferative and migratory cells that subsequently evade the treatment zone. Attempts to combine tumor surgery, radiation therapy, and chemotherapy have only managed to slow tumor growth temporarily, often resulting in drug resistance and therapy complications.”

3.       The 4th paragraph at row 54 and the 3rd paragraph at ow 43 could be merged, since they are strongly connected.

4.       The authors may find helpful to consider a recent paper on anticancer designed peptides that show promising performance: Prediction of Amphiphilic Cell-Penetrating Peptide Building Blocks from Protein-Derived Amino Acid Sequences for Engineering of Drug Delivery Nanoassemblies, J. Phys. Chem. B 2020, 124, 20, 4069–4078

Comments on the Quality of English Language

The English language needs extensive editing to become easier for reading.

Author Response

Dear reviewer,

Thank you very much for taking the time to review this manuscript.I am grateful for your response and deeply appreciate your suggestions for improving the article.  Your edits do indeed reflect much more accurately the thoughts we wanted to express in the course of our work. We will definitely use your suggestions in the process of correcting the article.

Reviewer 4 Report

Comments and Suggestions for Authors

Pavlova and colleagues investigated the potential effects of an aptamer-based treatment on glioma cells in culture, focusing on their impact on cell proliferation and migration. Given the challenges posed by the infiltrative nature of high-grade gliomas into the normal brain and the limited curative potential of surgical removal, the study explores a novel avenue that could enhance life quality and survival time. The manuscript presents a captivating finding—the Bi-(AID-1-T) G-quadruplex not only exhibits welcome anti-proliferative effects but surprisingly also demonstrates unwanted pro-migratory effects.

While the study offers valuable insights into this evolving field, there are a few areas that merit attention:

1.     Title and Terminology Consistency: The term "Janus" is confined to the title and is absent in the text. It might enhance clarity and cohesion to incorporate it at least once in the body of the manuscript or consider a title adjustment.

2.     Glioblastoma vs. Glioma Consistency: There is a slight inconsistency in the use of "glioblastoma" and "glioma" throughout the manuscript, e.g. Line 21: “glioblastoma”, but Line 22 refers to “glioma cell cultures (Grade III, IV)” – but that makes Line 21 wrong. Since glioblastoma is always WHO grade IV, but glioma can be I, II, III, or IV, the reviewer suggests checking and correcting this for clarity.

3.     Rare Events and References: Line 35 mentions the lack of gliomas forming distant metastases. It may be worthwhile to cite it as an extremely rare event, but note Sullivan et al.'s 2014 findings of glioblastoma cells in the bloodstream ( https://pubmed.ncbi.nlm.nih.gov/25139148/ ).

4.     Table 1 Clarity: In Table 1, under "Grade," the inclusion of III, IV, IV-, III-IV requires clarification. Discussing the evaluation process or confirming tumors by the same pathologist could address this. Especially the term “IV –“ is unclear, is it worse than “IV”. Additionally, replacing "IDH1-" with "wild-type" in the "IDH1 status" column could enhance readability.

5.     Inclusion of p53 Mutation Status: P53 is frequently mentioned, but the status of p53 mutations in the investigated tumors is omitted. Considering the prevalence of p53 mutations in high-grade glioblastomas, addressing this gap would strengthen the manuscript (just an example, but no necessitxy to cite this; https://pubmed.ncbi.nlm.nih.gov/1349850/ )

6.     CD44 Study Mention: Given the multiple references to CD44, acknowledging some glioblastoma-related work on study on CD44 could provide additional context (example, no necessity to cite this explicitly: (https://pubmed.ncbi.nlm.nih.gov/8750182/ ).

7.     Limitations of the Study: The manuscript should explicitly discuss the limitations, including the small sample size, variability in glioma types and grades, and the absence of p53 mutation status verification. Acknowledging these limitations transparently will enhance the credibility of the study.

In conclusion, despite these points of consideration, the reviewer supports the publication of the manuscript, recognizing its potential to contribute valuable insights to the field of glioma research.

Comments on the Quality of English Language

needs some language editing; check for apostrophs.

Author Response

Dear reviewer,

I thank you for reading my article so thoroughly. I am sure that by correcting the aspects you have pointed out, I will make our article better. I have attentively read all your comments and suggestions and a detailed response to them is given below.

  • Title and Terminology Consistency:The term "Janus" is confined to the title and is absent in the text. It might enhance clarity and cohesion to incorporate it at least once in the body of the manuscript or consider a title adjustment.

Thanks for the comment, we will try to include a mention of this effect in the text of the article.

  • Glioblastoma vs. Glioma Consistency:There is a slight inconsistency in the use of "glioblastoma" and "glioma" throughout the manuscript, e.g. Line 21: “glioblastoma”, but Line 22 refers to “glioma cell cultures (Grade III, IV)” – but that makes Line 21 wrong. Since glioblastoma is always WHO grade IV, but glioma can be I, II, III, or IV, the reviewer suggests checking and correcting this for clarity.

Thank you for the comment, I will definitely correct these issues

  • Rare Events and References:Line 35 mentions the lack of gliomas forming distant metastases. It may be worthwhile to cite it as an extremely rare event, but note Sullivan et al.'s 2014 findings of glioblastoma cells in the bloodstream ( https://pubmed.ncbi.nlm.nih.gov/25139148/ ).

Mentions of glioma metastases are extremely rare and the evidence for their existence is quite controversial. The article you kindly provided a link to mentions that such metastatic cases are mostly mesenchymal and have a number of mutations different from the original tumor. Therefore, we still think it is more correct to speak specifically about the absence of glioma metastasis outside the central nervous system. We ask permission not to raise this issue in this article, since this article does not study metastasis, but analyzes another process - the migration of glioma cells.

  • Table 1 Clarity: In Table 1, under "Grade," the inclusion of III, IV, IV-, III-IV requires clarification. Discussing the evaluation process or confirming tumors by the same pathologist could address this. Especially the term “IV –“ is unclear, is it worse than “IV”. Additionally, replacing "IDH1-" with "wild-type" in the "IDH1 status" column could enhance readability.

Yes, you're right, the table looks better that way. We'll replace this abbreviation

  • Inclusion of p53 Mutation Status:P53 is frequently mentioned, but the status of p53 mutations in the investigated tumors is omitted. Considering the prevalence of p53 mutations in high-grade glioblastomas, addressing this gap would strengthen the manuscript (just an example, but no necessitxy to cite this; https://pubmed.ncbi.nlm.nih.gov/1349850/ )

Yes, you are right, p53 mutation status is often included in tumor descriptions and is considered an important prognostic feature. However, we did not consider this in this work, as well as we did not consider the influence of these mutations on the behavior of glioma cell cultures.

  • CD44 Study Mention:Given the multiple references to CD44, acknowledging some glioblastoma-related work on study on CD44 could provide additional context (example, no necessity to cite this explicitly: (https://pubmed.ncbi.nlm.nih.gov/8750182/ ).

Thank you for the link to this very interesting work. In this article we consider CD44 without splice variants, but we will try to take this into account in our future work.

  • Limitations of the Study:The manuscript should explicitly discuss the limitations, including the small sample size, variability in glioma types and grades, and the absence of p53 mutation status verification. Acknowledging these limitations transparently will enhance the credibility of the study.

I thank you for this comment, we will definitely add this to our article

Round 2

Reviewer 1 Report

Comments and Suggestions for Authors

I thank the authors for their careful and useful changes to this manuscript

Author Response

I thank the Reviewer for the response. It was very important to me.

Reviewer 2 Report

Comments and Suggestions for Authors

The authors answered most of my comment and concerns, however, i would appreciate seeing data on IDH1+ cells using an optimized dose of your quadruplex compund.

Comments on the Quality of English Language

n/a

Author Response

Dear Reviewer,
I thank you for your reply.
Regarding IDH1+ cultures, we are just starting to experiment with them, so we cannot provide any data on them at the moment. 
If you would like and would be interested, you can leave any contact information so we can send you our next article, in which we will continue the study of bi(AID-1-T) using IDH1+ cultures, for your familiarization or review.